# Preliminary Evaluation Salivary Biomarkers in Patients with Oral Potentially Malignant Disorders (OPMD): A Case–Control Study

**DOI:** 10.3390/cancers15215256

**Published:** 2023-11-01

**Authors:** Pia López-Jornet, Aitana Olmo-Monedero, Camila Peres-Rubio, Eduardo Pons-Fuster, Asta Tvarijonaviciute

**Affiliations:** 1Faculty of Medicine and Odontology, Hospital Morales Meseguer, Clínica Odontológica, Marqués del Los Vélez s/n, 30008 Murcia, Spain; aitanaolmom@gmail.com; 2Interdisciplinary Laboratory of Clinical Analysis (Interlab-UMU), Veterinary School, Regional Campus of International Excellence Mare Nostrum, University of Murcia, Campus de Espinardo s/n, Espinardo, 30100 Murcia, Spain; camila.peres@um.es (C.P.-R.); asta@um.es (A.T.); 3Departamento de Anatomía Humana y Psicobiología, Faculty of Medicine and Odontology, Biomedical Research Institute (IMIB-Arrixaca), University of Murcia Spain, 30100 Murcia, Spain; eduardo.p.f@um.es

**Keywords:** OPMD, saliva, leukoplakia, lichen planus, ferritin, adenosine deaminase

## Abstract

**Simple Summary:**

Early diagnosis of potentially malignant oral disorders (OPMD) is crucial in reducing oral cancer mortality. It is necessary to investigate and identify predictive biomarkers capable of estimating the risk of malignant transformation in potentially malignant oral lesions. In this study, we analyzed a panel of salivary markers, including adenosine deaminase (ADA), ferritin (FRR) and total protein (TP). We found no significant differences in salivary ADA between the OPMD group and controls. However, we did observe statistically significant differences in ferritin and total proteins when compared to the control group. Therefore, ferritin and total proteins may serve as potential salivary biomarkers for diagnosis. Furthermore, saliva tests are a reliable and non-invasive diagnostic tool, offering an intriguing alternative for screening large populations. Nevertheless, validation through extensive clinical studies is required.

**Abstract:**

Introduction: Oral potentially malignant disorders (OPMD) are lesions associated with an increased risk of transformation (MT) into cancer. Objective: A study was made of the salivary levels of adenosine deaminase (ADA), ferritin (FRR) and total proteins (TP) in healthy individuals and in patients with oral potentially malignant disorders (OPMD), assessing the potential role of saliva as a diagnostic tool. Methods: A total of 91 subjects participated in the study, divided into two groups—59 patients with OPMD (oral leukoplakia or oral lichen planus) and 32 healthy controls—with measurements being made of salivary ADA, ferritin (FRR) and total proteins (TP). Results: There were no significant differences in salivary mean ADA between the OPMD group 0.85 ± 2.18 UI/I and the controls 0.71 ± 1.72 UI/I (*p* = 0.934), though the levels of both FRR mean OPMD, 12.66 ± 10.50 (µg/L), versus control, 7.19 ± 4.44 (*p* = 0.001), and TP, 23.41 ± 17, versus control, 14.15 ± 15.19, were significantly higher in the OPMD group (*p* = 0.001). Patients with oral lichen planus showed significant differences in terms of FRR (*p* = 0.009) and TP (*p* = 0.003). The ferritin in LPO with a cut-off point of 8.5C showed a sensitivity and specificity of 54.3% and 82.3, respectively. The area under the curve (AUC) was 0.69 (95% confidence interval (95% CI): 0.58–0.82; *p* = 0.003). Conclusions: Ferritin and total proteins may constitute potential salivary biomarkers for oral lichen planus, though further studies are still needed in this field. In addition, saliva testing is a reliable and noninvasive diagnostic tool and appears to be a reliable strategy offering an interesting alternative for the screening of large populations.

## 1. Introduction

Oral potentially malignant disorders (OPMD) are lesions associated with an increased risk of transformation (MT) into cancer. They constitute a heterogeneous group of conditions and often present as multiple lesions; careful clinical and pathological assessment is therefore required, with appropriate monitoring over time [1,2,3,4,5,6,7]. The early diagnosis and management of OPMD offers a unique opportunity for the development of strategies to prevent malignant transformation [5]. The 2020 World Health Organization (WHO) Collaborating Centers Consensus Workshop on OPMD updated the definition of OPMD as constituting clinical conditions that pose a risk of cancer development in the oral cavity arising from clinically definable precursor lesions or from clinically normal oral mucosa [1]. Thus, patients diagnosed with OPMD are more susceptible to developing oral cancer at some point in life [5]. Oral potentially malignant disorders comprise a heterogeneous group of conditions that include oral leukoplakia, oral lichen planus, proliferative verrucous leukoplakia, erythroplasia, oral submucous fibrosis, actinic cheilitis, oral lupus erythematosus, palatine lesions in reverse smokers, congenital dyskeratosis, lichenoid reactions and graft-versus-host disease (GVHD) [1,2,7]. The clinical manifestations of OPMD cover a broad range, with variations in color (white, red or a combination of both), texture (plaque, smooth, corrugated, verrucous, granular or atrophic) and size. In some cases, superficial microinvasive carcinoma may already be present and can be evidenced by histopathological study [1,7]. From the clinical perspective, the lesions may remain static or may either progress or regress over time [2]. On the other hand, OPMD can be located anywhere in the oral cavity, and may be present at one or more sites [7]. The malignant transformation rate varies throughout the world and is closely related to the type of OPMD and to other clinical, pathological or molecular factors, including lesion type, color, location and size, patient gender, and the presence and degree of dysplasia [1,2,7,8,9]. A recent systematic review [7] has reported a general malignant transformation rate in OPMD of 7.9% (99%CI 4.9–11.5%), though with high values in the heterogeneity test.

The early identification of suspicious oral lesions is very important in order to improve the outcome of the disease and reduce mortality [9]. However, opportunistic screening, based on simple, cost-effective, valid and reproducible techniques with minimum patient morbidity, remains a challenge [4,6]. There is a need to investigate and identify predictive biomarkers capable of estimating the risk of malignant transformation and allowing personalized and individualized OPMD management, with a view to helping reduce the incidence of oral cancer [9,10,11,12].

Adenosine deaminase is an enzyme that catalyzes the irreversible conversion of adenosine and/or deoxyadenosine into inosine and deoxyinosine, respectively. The enzyme plays an important role in detoxification processes and has also been shown to be necessary for monocyte differentiation into macrophages, and for the development of B and T lymphocytes [13,14,15]. Adenosine deaminase has been used as a cell-mediated immune and chronic inflammation marker. In saliva, the ADA levels have been seen to increase in squamous cell carcinoma of the tongue. Furthermore, Rai et al. [14] found that salivary ADA increases in squamous cell carcinoma as the disease progresses.

Ferritin is an iron storage protein related to acute and chronic inflammation. It appears to be over-expressed in patients with malignant tumors [16,17,18,19], and has been implicated in pathways associated with cancer, such as cell evasion and proliferation, angiogenesis, the inhibition of cell death, invasion and metastasis. However, few studies have evaluated ferritin in saliva in OPMD to date. The hypothesis of this study is the significant role that saliva can play in the diagnosis of potentially malignant oral disorders.

The aim was to evaluate the role of adenosine deaminase (ADA), ferritin (FRR) and total proteins (TP) in the fluid obtained after an oral rinse with sterile saline solution for half a minute in OPMD and controls.

## 2. Material and Methods

The present case–control study was designed according to the Strengthening Reporting of Observational Studies in Epidemiology (STROBE) guidelines. Salivary biomarkers in patients diagnosed with OPMD (oral lichen planus and oral leukoplakia) and healthy individuals were analyzed and compared.

### 2.1. Participants

The study was carried out in abidance with the principles of the Declaration of Helsinki and was approved by the Ethics Committee and Biosafety Committee of the University of Murcia (Murcia, Spain) (Reference: 4210/2022). Informed consent was obtained from all the participants in the study.

A general case history was compiled for each subject and an additional clinical intraoral examination was carried out, with entry of the data into a case report form designed for the study. The study was conducted from January 2022 to April 2023. A total of 91 individuals were consecutively enrolled from among the patients visiting the dental clinic of the Faculty of Medicine and Dentistry (Hospital Morales Meseguer, University of Murcia, Murcia, Spain) and who met the inclusion criteria detailed below.

### 2.2. Inclusion and Exclusion Criteria

The inclusion criteria for the OPMD group were patients with oral leukoplakia or oral lichen planus. Patients with clinical leukoplakia lesions (as defined by the WHO [1] “white plaques of questionable risk having excluded (other) known diseases or disorders that carry no increased risk for cancer”) were included.

To confirm the clinical diagnosis, a representative biopsy of the lesion was obtained in all cases for histopathological study, based on the established criteria [20,21].

The oral lichen planus clinical criteria included the following: the presence of bilateral, mostly symmetrical lesions; the presence of a lace-like network of slightly raised gray-white lines in a reticular pattern; and, by erosive, atrophic histopathology, (a) band-like zones of cellular infiltration confined to the superficial part of the connective tissue, (b) “liquefactive degeneration” in the basal layer, and (c) the absence of epithelial dysplasia.

The exclusion criteria were individuals under 18 years of age, patients with acute infectious processes, pregnant or nursing women, patients receiving medications that alter salivary secretion, subjects with previous neoplastic disease and subjected to head and neck radiotherapy and/or chemotherapy, anemic individuals and patients receiving iron supplements.

The control group, in turn, consisted of healthy individuals of similar age and habits that visited the dental clinic (Oral Medicine Unit) during the same period for other types of benign oral pathology, such as mucocele or benign tumors, without any sign or history OPMD (Figure 1 and Figure 2).

### 2.3. Data Collection

Information was obtained about alcohol use and smoking habits by self-administered questionnaires The following data were recorded in the OPMD group: gender, smoker status (smoker of >10 cigarettes/day, smoker of <10 cigarettes/day or non-smoker), alcohol intake (mild, moderate, severe or no alcohol), and characteristics of OPMD (gross appearance of the lesion, anatomical location and lesion size).

The following data were recorded in the control group: gender, race, date of birth, smoker status (smoker of >10 cigarettes/day, smoker of <10 cigarettes/day or non-smoker), alcohol intake (mild, moderate, severe or no alcohol).

### 2.4. Saliva Sampling

Saliva sampling was carried out in the morning between 9 a.m. and 11 a.m. in all cases. The patients received instructions prior to sampling (no smoking, drinking, eating or brushing of the teeth for at least 30 min before saliva collection). For sampling, the patients were required to perform an oral rinse with 10 mL of sterile saline solution (0.9% NaCl) contained in a 50 mL Falcon tube and were instructed to spit back into the same tube after 30 s. Samples containing visible blood traces were discarded. The collected saliva samples were transported in ice to the laboratory, where they were centrifuged (2600 rpm at 23 °C during 15 min) and frozen at −80 °C until analysis.

### 2.5. Biochemical Analysis

Total ADA was measured using a commercial automated spectrophotometric method (Adenosine Deaminase assay kit, Diazyme Laboratories, Poway, CA, USA). This method is based on the enzymatic deamination of adenosine to inosine, which is converted to hypoxanthine via purine nucleoside phosphorylase. The hypoxanthine is then converted to uric acid and hydrogen peroxide by xanthine oxidase. Peroxidase is further reacted with N-ethyl-N-(2-hydroxy-3-sulfopropyl)-3-methylaniline and 4-aminoantipyrine in the presence of peroxidase to generate quinine dye, which is kinetically monitored at a wavelength of 550 nm [13,22].

Ferritin in the saliva samples was measured using a commercial automated immunoturbidimetric assay (Ferritin latex, 31.935, Biosystems S.A., Barcelona, Spain). In turn, total protein content was measured colorimetrically with a commercial kit (Beckman Coulter, Brea, CA, USA, OSR6132), according to the instructions of the manufacturer.

### 2.6. Statistical Analysis

A descriptive analysis was made of the quantitative (mean, standard deviation [SD], minimum and maximum, median and quartiles) and qualitative variables (absolute frequencies and percentages). The salivary parameters followed a non-normal distribution in both the OPMD group and the control group, as evidenced by the Kolmogorov–Smirnov test. Since the sample size was modest in both groups (*n* = 59 and *n* = 32, respectively), a nonparametric statistical approach was used. The inferential analysis was based on the chi-square test, which measures the association between two categorical variables such as, for example, smoking and group, and the Mann–Whitney U test, which compares the distribution of values of a salivary marker in two independent groups (e.g., OPMD and control). The Bonferroni correction was used in the case of multiple comparisons. In turn, the Kruskal–Wallis test was used to compare the distribution of values of a salivary marker in more than two independent groups (control, oral lichen planus and leukoplakia). The Spearman correlation test was used to assess non-linear associations between variables. The level of statistical significance was established as 5% (α = 0.05). The SPSS version 18 statistical package was used throughout.

## 3. Results

A total of 91 subjects participated in the study, divided into two groups: 32 healthy controls and 59 patients with OPMD. The overall mean age was 64.5 ± 11.1 years (range 29–94), with a gender distribution of 74 females (81.3%) and 17 males (18.7%) (*p* = 0.001).

The sociodemographic data, clinical characteristics and habits of the study groups are shown in Table 1. The overall active smoker rate was 9.9%, and 42% of the subjects in the OPMD group had never smoked, versus 70% in the control group (*p* = 0.011).

Within the OPMD group, 46 patients were diagnosed with oral lichen planus (78.0%) and 11 with leukoplakia (18.6%). The diagnosis could not be confirmed in two cases, and these patients were excluded from the study analysis. The mean time from diagnosis was two years (interquartile range [IQR] 0–6 years).

### 3.1. Salivary Biomarkers according to Study Group

The findings referred to the salivary biomarkers are reported in Table 2. The analysis of amounts per unit volume corresponding to ADA, ferritin and total proteins revealed significant differences in ferritin and total proteins in the OPMD group versus the controls (*p* = 0.001), though statistical significance was not reached in the case of ADA. Likewise, on considering the different oral disorders, no significant differences were observed between the control group and oral lichen planus (*p* > 0.05), the control group and leukoplakia (*p* = 0.438) or between oral lichen planus and leukoplakia (*p* = 0.207).

In relation to oral lichen planus, significant differences were recorded for ferritin (*p* = 0.009) and total proteins (*p* = 0.003). However, no significant differences were observed in relation to oral leukoplakia for either of these analytes.

### 3.2. Biomarkers according to Independent Factors (Table 3)

In Table 3, we can observe the results obtained for the clinical characteristics of the sample (gender, age, time from diagnosis, type of lichen planus, location and size) and the analysis of the salivary biomarkers assessed, which include ferritin, ADA, and total proteins.

**Table 3 cancers-15-05256-t003:** Comparison of the biomarkers ADA, FRR and total proteins according to independent factors (Mann–Whitney U test [MW], Kruskal–Wallis test [KW] and Pearson correlation coefficient [r]).

	ADA	FRR	PROT
Gender	0.068 (MW)	0.256 (MW)	0.482 (MW)
Age	r = 0.12 (*p* = 0.254)	r = 0.13 (*p* = 0.235)	r = 0.14 (*p* = 0.182)
Smoking	0.231 (KW)	0.475 (KW)	0.399 (KW)
Alcohol	0.276 (KW)	0.176 (KW)	0.758 (KW)
Time from diagnosis	r = −0.11 (*p* = 0.935)	r = −0.11 (*p* = 0.395)	r = 0.06 (*p* = 0.637)
Lichen—striated	0.491 (MW)	0.927 (MW)	0.396 (MW)
Lichen—erosive	0.161 (MW)	0.317 (MW)	0.111 (MW)
Location: cheek mucosal	0.994 (MW)	0.767 (MW)	0.064 (MW)
Location: gingival mucosa	0.944 (MW)	0.902 (MW)	0.934 (MW)
Location: lip mucosa/buccal sulcus	0.947 (MW)	0.664 (MW)	0.888 (MW)
Location: tongue	0.504 (MW)	0.602 (MW)	0.884 (MW)
Location: other zones	0.947 (MW)	0.613 (MW)	0.431 (MW)
Size	0.048 * (MW)	0.667 (MW)	0.045 * (MW)

* *p* < 0.05.

### 3.3. Correlations between Biomarkers

Lastly, an analysis was made to explore possible correlations between the three analyzed biomarkers (ADA, FRR and total proteins). Table 4 shows most of the correlations to be significant in both the global sample and in each of the groups.

### 3.4. The Sensitivity and Specificity of Salivary Biomarkers in Patients with Oral Lichen Planus

The Ferritin in LPO with a cut-off point of 8.5C showed a sensitivity and specificity of 54.3% and 82.3, respectively. The area under the curve (AUC) was 0.69 (95% confidence interval (95% CI): 0.58–0.82; *p* = 0.003) (Figure 3A).

The PT in LPO with a cut-off point of 10.7 showed a sensitivity and specificity of 84.8% and 56.3%, respectively. The area under the curve (AUC) was found to be 0.72 (95% confidence interval (95% CI): 0.58–0.82; *p* = 0.001) (Figure 3B).

## 4. Discussion

The identification of biomarkers is essential in the development of techniques for the early diagnosis and management of OPMD, particularly in populations at risk. The diagnostic potential of a number of candidate biomarkers has been investigated over the last decade, though it is doubtful that any single biomarker will be able to invariably recognize oral squamous cell carcinoma, due to the many mechanisms involved in carcinogenesis, tumor heterogeneity and substantial variation of the risk factors [9,10]. Cancer is one of the leading causes of death worldwide, and advances in its treatment, early detection and prevention have helped to reduce its impact. Studies in the various animal models used for research in recent years are essential to observe the progression of oral cancer from early to advanced stages, and this is essential for understanding of how oral cancer develops and spreads, for identification of biomarkers of the disease and to guide the therapeutic approach [23,24].

It is thus more likely that combinations of biomarkers will be able to afford greater diagnostic reliability [25,26,27]. In the present study we therefore analyzed a panel of markers in the form of salivary adenosine deaminase (ADA), ferritin (FRR) and total proteins (TP).

Adenosine deaminase (ADA) is a protein produced by the cells of the body that is related to the activation of lymphocytes, which, in turn, play a key role in the immune response [13]. Some authors [28,29,30] have reported significantly increased serum levels of ADA in patients with head and neck cancer, thus suggesting that serum ADA may be useful for the diagnosis and follow-up of the disease. Rai et at. [14] evaluated the activity of salivary ADA in individuals with tongue cancer, and observed differences with respect to the salivary ADA concentrations in healthy subjects (*p* < 0.001). In addition, the serum ADA levels were seen to increase significantly with the progression of squamous cell carcinoma of the tongue from stage I to stage III. In contrast, Saracoglu et al. [31] analyzed the activity of salivary ADA and 5’-nucleotidase in oral and laryngeal cancer and recorded no such significant differences. Likewise, in the present study, we found no significant differences in salivary ADA between the OPMD group and the controls, and there were also no differences on comparing the concrete diagnoses of oral lichen planus and oral leukoplakia, thus underscoring the need for further research in this field—the diagnostic and prognostic value of such markers often requires validation through extensive clinical studies.

Another aspect which we explored in our study is in relation to ferritin and total proteins. The evidence points to a clear role for iron in oral carcinogenesis, and in this regard, ferritin has been investigated due to its function as the main iron-storing protein in cells. It has been shown that ferritin is a multifunctional protein with implications in cell proliferation, angiogenesis, immunosuppression and iron supply. In the context of cancer, the ferritin levels are seen to be increased in the serum of many cancer patients, and the highest levels are correlated to more aggressive disease and a poor clinical outcome. Moreover, recent research has identified ferritin as playing a key role in tumor progression and resistance to therapy [16,17,18,19]. Elevated ferritin levels have been reported in head and neck cancer [32,33,34,35,36]. In this respect, Bhatavdekar et al. [35] observed elevated ferritin in patients with head and neck squamous cell carcinoma (HNSCC). After 8 months of treatment, the levels returned close to normal, though in patients with a poor prognosis the levels tended to increase or remain high. Ferritin, therefore, may constitute a useful tool in head and neck malignancies. In 2019, Hu et al. [36] established a correlation between ferritin expression levels and neck metastasis in head and neck cancer, using it to predict metastasis in the neck lymph nodes in patients with HNSCC.

In relation to OPMD, we observed statistical differences in ferritin and total proteins versus the control group. This is consistent with the findings of Wu et al. [37,38], who recorded significantly higher levels of carcinoembryonic antigen (CEA), squamous cell carcinoma antigen (SCC-Ag) and ferritin in the serum of patients with oral precancer than in the healthy controls. Thus, serum CEA, SCC-Ag and ferritin may represent potential tumor markers for the detection of patients with oral precancerous lesions [37,38].

Buch et al. [39] analyzed correlations between ferritin levels in saliva and serum in healthy individuals, subjects with OPMD and oral cancer patients. All three groups showed a significant positive correlation between the ferritin levels in saliva and serum: healthy individuals (r = 0.622), subjects with OPMD (r = 0.878) and oral cancer patients (r = 0.668). Furthermore, the study highlighted saliva as a reliable and noninvasive diagnostic tool for the screening of large populations.

In a recent study, Wu and Chiang [40] documented an increase in serum ferritin in patients with lichen planus. In line with the above authors, our study showed the ferritin and total protein levels to be significantly higher in patients with oral lichen planus than among the controls.

Our study has limitations, starting with its cross-sectional design. In this regard, longitudinal studies involving longer follow-up periods would be advisable. On the other hand, this is a preliminary study that will require an increase in sample size. Lastly, it must be mentioned that the type and quality of the diet of the patients, and their body mass index, were not taken into account. It should be mentioned or reiterated that the type of statistical tests done were nonparametric and that this can weaken the evidence presented. As mentioned, OPMD is an umbrella term for a heterogenous group.

In this regard, the investigation must be continued, seeking markers capable of predicting malignant transformation while always taking into account the great heterogeneity of the molecular profiles and the differences in evolution among the individuals. Saliva is a candidate biological fluid for the analysis and identification of biomarkers. Detection is advantageous in this respect, since saliva sampling is easy, noninvasive and better tolerated by patients, reducing the anxiety and discomfort associated with blood sampling, and increasing patient willingness to undergo frequent controls [10,13]. On the other hand, issues related to standardization, processing and analysis of the saliva samples must be addressed so that such markers can be used as additional tools in the diagnosis and prognosis of OPMD.

## 5. Conclusions

Ferritin and total proteins may constitute potential salivary biomarkers for oral lichen planus, though further studies are still needed in this field. In addition, saliva testing is a reliable and noninvasive diagnostic tool and appears to be a reliable strategy offering an interesting alternative for the screening of large populations.

## Figures and Tables

**Figure 1 cancers-15-05256-f001:**
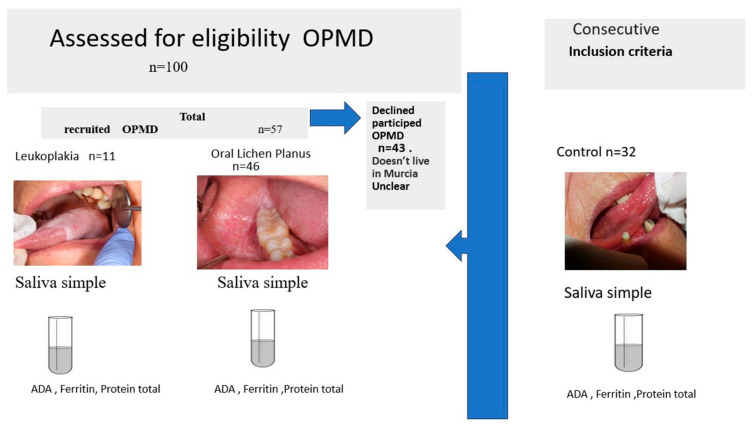
Study Design.

**Figure 2 cancers-15-05256-f002:**
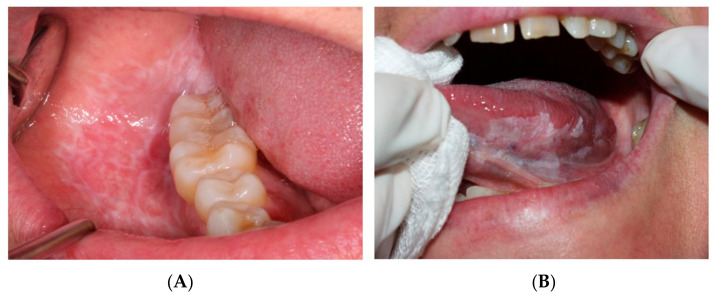
Potentially malignant oral lesions. (**A**) Oral lichen planus on the buccal mucosa. (**B**) Oral leukoplakia on the ventral surface of the tongue.

**Figure 3 cancers-15-05256-f003:**
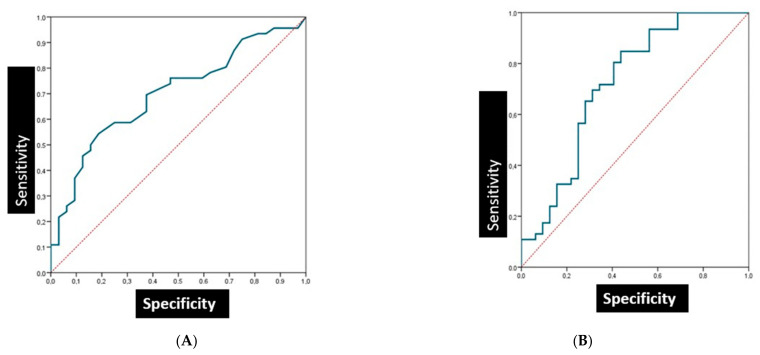
Analysis of salivary biomarkers. (**A**) Ferritin oral lichen planus. (**B**) Protein total oral lichen planus.

**Table 1 cancers-15-05256-t001:** Sociodemographic data, clinical characteristics and habits of the study groups according to factors (Mann–Whitney U test [MW] and Chi^2^).

Variables		CONTROL		OPMD		*p* Value
		N	%	N	%	
**Age** **Mean ± SD**		63.6 ± 8.8		65.0 ± 12.2		>0.05
	Yes, <10 cig/day	0	0	3	5.1	0.011
	Ex-smoker	7	21.9	27	45.8	
	No	23	71.9	25	42.4	
Alcohol	Yes, severe	0	0	3	5.1	
	Yes, moderate	4	12.5	10	16.9	>0.05
	Yes, mild	9	20.1	19	32.2	
	No	19	59.4	27	45.8	
Type of lesion	Oral lichen planus			46	80.7	
	Leukoplakia	-	-	11	19.3	
Lichen planus	Reticular	-	-	30	65.2	
	Erosive	-	-	16	34.8	
Location	Cheek mucosa	-	-	30	50.8	
	Gingival mucosa	-	-	18	30.5	
	Lip	-	-	10	16.9	
	Tongue	-	-	13	22	
	other zones	-	-	16	27.1	
Lesion size	<2 cm	-	-	18	30.5	
	>2 cm	-	-	41	69.5	

**Table 2 cancers-15-05256-t002:** Salivary biomarkers according to study group.

		Total (*n* = 89)	Control (*n* = 32)	OPMD (*n* = 57)	*p*-Value
ADA (IU/L)	Mean ± SD	0.80 ± 2.02	0.71 ± 1.72	0.85 ± 2.18	0.934
	Median (IQR)	0.28 (0.2–14.9)	0.30 (0.10–0.57)	0.28 (0.11–0.77)	
FRR (µg/L)	Mean ± SD	10.74 ± 9.20	7.19 ± 4.44	12.66 ± 10.50	0.001
	Median (IQR)	9.40 (2.5–53.9)	6.20 (4.50–8.15)	9.40 (6.50–14.50)	
Proteins (mg/dL)	Mean ± SD	20.16 ± 17.44	14.15 ± 15.19	23.41 ± 17.83	0.001
	Median (IQR)	14.8 (4–86.1)	9.9 (3.0–21.0)	18.9 (11.5–27.4)	

**Table 4 cancers-15-05256-t004:** Comparison of the biomarkers ADA, FRR and total proteins according to independent factors.

	Total Sample	Control	OPMD
ADA vs. FRR	r = 0.37 (*p* < 0.001 ***)	r = 0.15 (*p* = 0.419)	r = 0.52 (*p* < 0.001 ***)
ADA vs. PROT	r = 0.53 (*p* < 0.001 ***)	r = 0.61 (*p* < 0.001 ***)	r = 0.53 (*p* < 0.001 ***)
FRR vs. PROT	r = 0.44 (*p* < 0.001 ***)	r = 0.41 (*p* = 0.020 *)	r = 0.35 (*p* = 0.007 **)

* *p* < 0.05; ** *p* < 0.01; *** *p* < 0.001.

## Data Availability

All data are available as part of this manuscript. There are no additional data that are not included in the tables published within the text of the manuscript.

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
