# Peer review of "Preliminary Evaluation Salivary Biomarkers in Patients with Oral Potentially Malignant Disorders (OPMD): A Case–Control Study"

_cancers, 2023, doi:10.3390/cancers15215256_

Round 1
Reviewer 1 Report
Comments and Suggestions for Authors
Dear,
Authors investigated the the salivary levels of adenosine deaminase (ADA), ferritin (FRR) and total proteins in healthy individuals and in patients with oral potentially malignant disorders (OPMD), assessing the potential role of saliva as a diagnostic tool. There are some comments which should be followed;
-Title should be revised and summarized
- There are some error English grammar in whole text
-Introduction is poor about cancer explanation and I suggest the below references:
*Khayatan D, Hussain A, Tebyaniyan H. Exploring animal models in oral cancer research and clinical intervention: A critical review. Vet Med Sci. 2023.
*Mosaddad SA, Beigi K, Doroodizadeh T, Haghnegahdar M, Golfeshan F, Ranjbar R, et al. Therapeutic applications of herbal/synthetic/bio-drug in oral cancer: An update. Eur J Pharmacol. 2021;890:173657.
-Method and materials are well-organized
-Results are well-explained.
-Discussion and conclusion are ok.
Author Response
Referee
We welcome all helpful suggestions for improving this manuscript. We have modified the manuscript to directly address all reviewer comments. The modifications are detailed in the document
Title should be revised and summarized OK
- There are some error English grammar in whole text
-Introduction is poor about cancer explanation and I suggest the below references:
Has been added
Khayatan D, Hussain A, Tebyaniyan H. Exploring animal models in oral cancer research and clinical intervention: A critical review. Vet Med Sci. 2023 Jul;9(4):1833-1847. doi: 10.1002/vms3.1161. Epub 2023 May 17. PMID: 37196179
Mosaddad SA, Beigi K, Doroodizadeh T, Haghnegahdar M, Golfeshan F, Ranjbar R, Tebyanian H. Therapeutic applications of herbal/synthetic/bio-drug in oral cancer: An update. Eur J Pharmacol. 2021 Jan 5;890:173657. doi: 10.1016/j.ejphar.2020.173657.
Thank you for the comment
Reviewer 2 Report
Comments and Suggestions for Authors
The manuscript by López Jornet et al. aimed to study the salivary levels of adenosine deaminase (ADA), ferritin (FRR), and total proteins in healthy individuals and in patients with oral potentially malignant disorders (OPMD), assessing the potential role of saliva as a diagnostic tool. The authors were able to find differences in ADA, FRR, and total proteins between the groups. However, the data does not demonstrate that these biomolecules could be used as a diagnostic tool for OPMD. In order to prove that a different statistical analysis must be performed, such as a ROC curve. In addition, since oral leukoplakia and oral lichen planus are different lesions, with different risk levels of malignant transformation, it does not make sense to analyze them together. The authors should re-evaluate their claims based on the data that they have.
Comments on the Quality of English LanguageThe language should be edited by a native speaker. Grammar errors could be found throughout the manuscript.
Author Response
We welcome all helpful suggestions for improving this manuscript. We have modified the manuscript to directly address all reviewer comments. The modifications are detailed in the document Thank you for the comment
In order to prove that a different statistical analysis must be performed, such as a ROC curve. In addition, since oral leukoplakia and oral lichen planus are different lesions, with different risk levels of malignant transformation, it does not make sense to analyze them together. The authors should re-evaluate their claims based on the data that they have.
I agree in medical research, the assessment of biomarkers and their potential clinical applications is an ongoing and complex process. Heterogeneity: OPMDs can encompass various conditions with different underlying molecular mechanisms. .We have performed the indicated statistical ROC curve analysis. It was only significant in lichen planus not in leukoplakia, we proceeded to the analysis of the Roc curves.
The biomolecule in question might not be universally applicable to all types of OPMDs, Oral Potentially Malignant Disorders can be complex diseases influenced by multiple factors. Determining which OPMD will follow a stable clinical course and which will progress to invasive carcinoma is challenging with routine histopathologic diagnosis and has limited prognostic value. Therefore, the development of alternative methods to predict the malignant potential of suspicious lesions is much needed.
We have performed the indicated statistical ROC curve analysis. It was only significant in lichen planus not in leukoplakia, we proceeded to the analysis of the Roc curves.
The sensitivity and specificity of salivary biomarkers in patients with oral lichen planus we found.
The Ferritin in LPO with a cut-off point of 8.5C showed a sensitivity and specificity of 54.3% and 82.3, respectively The area under the curve (AUC) iwas 0.69 (95% confidence interval (95% CI]: 0.58–0.82; p = 0.003) (Figure 3A).
The PT in LPO with a cut-off point of 10.7 showed a sensitivity and specificity of 84.8% and 56.3%respectivelyThe area under the curve (AUC) was found to be 0.72 (95% confidence interval (95% CI): 0.58–0.82; p = 0.001) (Figure 3B).
Reviewer 3 Report
Comments and Suggestions for Authors
Thank you for granting me the opportunity to review this manuscript, which aims to investigate salivary levels of adenosine deaminase (ADA), ferritin (FRR), and total proteins in both healthy individuals and patients with oral potentially malignant disorders. The study seeks to evaluate the potential of saliva as a diagnostic tool in this context.
While the topic of the manuscript is intriguing, there are certain deficiencies that need to be addressed in order to enhance its quality.
Recommended Changes:
Abstract section - Place angle brackets before the "Result" subsection.
Abstract section, results - In addition to reporting p-values, include the actual values of adenosine deaminase (ADA), ferritin (FRR), and total proteins.
Abstract - Craft a more specific conclusion based on the findings obtained.
Introduction - Present the study's hypothesis towards the end of the introduction.
Methodology - Please calculate the study's statistical power or justify the sample size chosen.
Methodology - Address the poor resolution of Figure 1. Enhance the resolution or consider removing the image.
Results - Clarify the meaning of the second variable labeled as "tuxedo" in tbl1. Specify the statistical methods used in the legend at the bottom of the table.
Maintain consistent capitalization for "p value" and "P value" throughout the manuscript.
Include the median and interquartile range (IQR) for the "total" variable in table two.
Consider utilizing correlation or regression analyses for the content of Table 3, to enhance its quality.
Reevaluate the information presented in tbl 4 under the category "total sum." Provide a comparison between "control" and "OPMD" using the Mann-Whitney test.
In the discussion, provide more detailed commentary and explanation of the results, avoiding redundant statements that replicate the introduction.
Strengthen the conclusion by making it more specific and directly tied to the obtained results.
Revise the literature review to align with the specific instructions of the Journal.
Addressing these changes will significantly improve the manuscript and align it better with the standards expected by the Journal.
Author Response
We welcome all helpful suggestions for improving this manuscript. We have modified the manuscript to directly address all reviewer comments. The modifications are detailed in the document
We have corrected typographical errors; Place angle brackets before the "Result" subsection. OK - In addition to reporting p-values, include the actual values of adenosine deaminase (ADA), ferritin (FRR), and total proteins. Has been added conclusion based on the findings obtained. Has been added
Present the study's hypothesis towards the end of the introduction. has been included
We have improved the quality of the image
We have Revised the literature o align with the specific instructions of the Journal.
Reviewer 4 Report
Comments and Suggestions for Authors
This article was described some important information.
However, some points need to be improvement.
1, The authors should show more clear images that representative of cases of Leukoplakia and Lichen Planus. And Histopathological images need too.
2, Please add more detail criteria for histopathological diagnosis of Leukoplakia, eg: Acanthosis, Hyperkeratosis, parakeratosis and so on. Because Leukoplakia is not definitive diagnosis, it is clinical diagnosis.
Author Response
We welcome all helpful suggestions for improving this manuscript. We have modified the manuscript to directly address all reviewer comments. The modifications are detailed in the document
1, The authors should show more clear images that representative of cases of Leukoplakia and Lichen Planus. And Histopathological images need too. Has been added
2, Please add more detail criteria for histopathological diagnosis of Leukoplakia, eg: Acanthosis, Hyperkeratosis, parakeratosis and so on. Because Leukoplakia is not definitive diagnosis, it is clinical diagnosis has been added
The following criteria should be considered when making a clinical diagnosis of oral leukoplakia:
A predominantly white patch/plaque that cannot be rubbed off.
Most homogeneous Leukoplakia affect a circumscribed area and have well-demarcated borders. A smaller subset can present with diffuse borders.
Non-homogeneous Leukoplakia typically present with more diffuse borders and may have red or nodular components.
No evidence of chronic traumatic irritation to the area (e.g., a sharp tooth rubbing on the tongue, a white patch on the alveolar ridge or retromolar pad from masticatory friction, a white patch on gingiva from overzealous toothbrushing).
Is not reversible on elimination of apparent traumatic causes, that is demonstrates a persistence feature.
Does not disappear or fade away on stretching (retracting) the tissue.
According to the World Health Organization (WHO), lesions and conditions of the oral mucosa predisposed to malignant conversion are defined as OPMD. Typical representatives are oral leukoplakia, erythroplakia, lichen planus and submucosal fibrosis
van der Waal I. Oral leukoplakia: A diagnostic challenge for clinicians and pathologists. Oral Dis. 2019;25:348–349
Round 2
Reviewer 1 Report
Comments and Suggestions for Authors
Dear, the revised version is acceptable.
Author Response
Many thanks for your kind notice
Reviewer 2 Report
Comments and Suggestions for Authors
Thank you very much for the corrections. Your manuscript has improved substantially. However, your abstract must be accurate. Some of the results in your manuscript are not included, and the conclusion in your abstract differs from the one at the end of the manuscript.
Author Response
Thank you very much for the corrections. I have addressed the comments from reviewer 2 and have made the necessary changes. Your manuscript has improved substantially. Thank you However, I would like to point out that I have revised the abstract to ensure accuracy and included the missing results in the manuscript. Has been added The conclusion in the abstract now aligns with the one at the end of the manuscript." Has been changesReviewer 3 Report
Comments and Suggestions for Authors
Thank you to the authors for accepting individual corrections. However, in order to change my stance, it is necessary to correct everything that is required. I would recommend to the authors that in the future, they respond to each reviewer's request in greater detail, making it easier to follow, rather than providing only general responses in a few sentences.
Author Response
Reviewer 3
Response Many thanks for your kind notice and review. Changed as indicated throughout the text. We welcome all helpful suggestions for improving this manuscript. We have modified the manuscript to directly address all reviewer comments. The modifications are detailed in the document
. Abstract section –
- Place angle brackets before the "Result" subsection.
- Abstract section, results - In addition to reporting p-values, include the actual values of adenosine deaminase (ADA), ferritin (FRR), and total proteins.
- Abstract - Craft a more specific conclusion based on the findings obtained.
- We have corrected typographical errors; Place angle brackets before the "Result" subsection.
OK - In addition to reporting p-values, include the actual values of adenosine deaminase (ADA), ferritin (FRR), and total proteins. Has been added conclusion based on the findings obtained. Has been added
Introduction - Present the study's hypothesis towards the end of the introduction.
Response In this study, we utilize reject the null hypothesis of no potential role effect of saliva as a diagnostic tool in OPMD.
Methodology –
Please calculate the study's statistical power or justify the sample size chosen.
Methodology - Address the poor resolution of Figure 1. Enhance the resolution or consider removing the image.
I agree section was added
Results
- Clarify the meaning of the second variable labeled as "tuxedo" in tbl1. Specify the statistical methods used in the legend at the bottom of the table.
- Maintain consistent capitalization for "p value" and "P value" throughout the manuscript.
- Include the median and interquartile range (IQR) for the "total" variable in table two.
- Consider utilizing correlation or regression analyses for the content of Table 3, to enhance its quality.
- Reevaluate the information presented in tbl 4 under the category "total sum." Provide a comparison between "control" and "OPMD" using the Mann-Whitney test.
I agree section was added. Changed as indicated, the previous draft did not clearly state
In the discussion
Provide more detailed commentary and explanation of the results, avoiding redundant statements that replicate the introduction.
Strengthen the conclusion by making it more specific and directly tied to the obtained results.
Many thanks Has been modified
References
Revise the literature review to align with the specific instructions of the Journal.
Addressing these changes will significantly improve the manuscript and align it better with the standards expected by the Journal.
Many thanks References have been reviewed according to the journal's standards
Response Many thanks for your kind notice and review. Changed as indicated throughout the text. We welcome all helpful suggestions for improving this manuscript. We have modified the manuscript to directly address all reviewer comments. The modifications are detailed in the document
. Abstract section –
- Place angle brackets before the "Result" subsection.
- Abstract section, results - In addition to reporting p-values, include the actual values of adenosine deaminase (ADA), ferritin (FRR), and total proteins.
- Abstract - Craft a more specific conclusion based on the findings obtained.
- We have corrected typographical errors; Place angle brackets before the "Result" subsection.
OK - In addition to reporting p-values, include the actual values of adenosine deaminase (ADA), ferritin (FRR), and total proteins. Has been added conclusion based on the findings obtained. Has been added
Introduction - Present the study's hypothesis towards the end of the introduction.
Response In this study, we utilize reject the null hypothesis of no potential role effect of saliva as a diagnostic tool in OPMD.
Methodology –
Please calculate the study's statistical power or justify the sample size chosen.
Methodology - Address the poor resolution of Figure 1. Enhance the resolution or consider removing the image.
I agree section was added
Results
- Clarify the meaning of the second variable labeled as "tuxedo" in tbl1. Specify the statistical methods used in the legend at the bottom of the table.
- Maintain consistent capitalization for "p value" and "P value" throughout the manuscript.
- Include the median and interquartile range (IQR) for the "total" variable in table two.
- Consider utilizing correlation or regression analyses for the content of Table 3, to enhance its quality.
- Reevaluate the information presented in tbl 4 under the category "total sum." Provide a comparison between "control" and "OPMD" using the Mann-Whitney test.
I agree section was added. Changed as indicated, the previous draft did not clearly state
In the discussion
Provide more detailed commentary and explanation of the results, avoiding redundant statements that replicate the introduction.
Strengthen the conclusion by making it more specific and directly tied to the obtained results.
Many thanks Has been modified
References
Revise the literature review to align with the specific instructions of the Journal.
Addressing these changes will significantly improve the manuscript and align it better with the standards expected by the Journal.
Many thanks References have been reviewed according to the journal's standards
Round 3
Reviewer 3 Report
Comments and Suggestions for Authors
Dear authors,
Thanks for accepting requested midifications but I am sill of opinion that discussion section saund more like Introduction. You should only discussion your results.
Author Response
Thank you for taking the time to review our manuscript and for providing valuable feedback. We appreciate your efforts in helping us improve the clarity and structure of our paper.
We have carefully considered your comment regarding the discussion section of our manuscriptIn the discussion section we have contextualized and analyzed our study results; Interpretation of Results. Comparison with existing Literature, Revisit the research questions you initially posed in the introduction, Implications, Limitations and And future research Reiterate the importance and relevance of your research
Round 4
Reviewer 3 Report
Comments and Suggestions for Authors
Introduction - Present the study's hypothesis towards the end of the introduction.
The authors state "On the other 2hand, this is a preliminary study that will require an increase in sample size." - then mention the same in the title.
Include the median and interquartile range (IQR) for the "total" variable in table two.
What does 1, 2, 3 stars mean in tbl4.
References not corrected.
The authors ask me to repeat the same review 4 times. Let them correct everything that has been requested so far.
Author Response
We greatly appreciate your constructive comments and suggestions, which have been extremely helpful in improving the quality and clarity of the work. His knowledge and experience have been fundamental in this process and have contributed significantly to the quality of the research.
We are pleased to inform you that we have carefully implemented all suggested revisions and changes to the manuscript .
Thank you for the feedback. Here are the responses to your comments:
Introduction: The study's hypothesis will be presented towards the end of the introduction, as suggested. OK
The authors' statement regarding the need for an increase in sample size will be incorporated into the title, as requested. Has been added
We will include the median and interquartile range (IQR) for the "total" variable in table two. OK
The meaning of the 1, 2, and 3 stars in tbl4 will be clarified in the table or its accompanying legend for better comprehension. OK
The references will be reviewed and corrected. OK